# Genomic and functional determinants of host spectrum in Group B *Streptococcus*

Chiara Crestani[1¤a]*, Taya L. Forde[1¤b], John Bell[2], Samantha J. Lycett[3], Laura M. A. Oliveira[4], Tatiana C. A. Pinto[4], Claudia G. Cobo-Ángel[5¤c], Alejandro Ceballos-Márquez[5], Nguyen N. Phuoc[6], Wanna Sirimanapong[7], Swaine L. Chen[8,9], Dorota Jamrozy[10], Stephen D. Bentley[10], Michael Fontaine[2¤d], Ruth N. Zadoks[1,2,11]

**1** Institute of Biodiversity, Animal Health & Comparative Medicine, University of Glasgow, Glasgow, Scotland, United Kingdom, **2** Moredun Research Institute, Penicuik, Scotland, United Kingdom, **3** The Roslin Institute, University of Edinburgh, Midlothian, Scotland, United Kingdom, **4** Instituto de Microbiologia Paulo de Goes, Federal University of Rio de Janeiro, Rio de Janeiro, State of Rio de Janeiro, Brazil, **5** CLEV research group, Universidad de Caldas, Manizales, Caldas, Colombia, **6** Faculty of Fisheries, University of Agriculture and Forestry, Hue University, Hue, Vietnam, **7** Faculty of Veterinary Science, Mahidol University, Nakhon Pathom, Thailand, **8** Infectious Diseases Translational Research Programme, Department of Medicine, Division of Infectious Diseases, Yong Loo Lin School of Medicine, National University of Singapore, Singapore, **9** Laboratory of Bacterial Genomics, Genome Institute of Singapore, Singapore, **10** Parasites and Microbes Programme, Wellcome Sanger Institute, Hinxton, England, United Kingdom, **11** Sydney School of Veterinary Science, Faculty of Science, University of Sydney, Camden, NSW, Australia

¤a Current address: Department of Global Health, Institut Pasteur, Paris, France
¤b Current address: School of Biodiversity, One Health & Veterinary Medicine, University of Glasgow, Glasgow, Scotland, United Kingdom
¤c Current address: College of Veterinary Medicine, Cornell University, Ithaca, NY, United States of America
¤d Current address: Point Horizon Ltd, Livingston, Scotland, United Kingdom
* chiara.crestani@pasteur.fr

**Data Availability Statement:** All raw or assembled Illumina sequence data is available from the European Nucleotide Archive (ENA) or from the National Center for Biotechnology Information

## Abstract

Group B *Streptococcus* (GBS) is a major human and animal pathogen that threatens public health and food security. Spill-over and spill-back between host species is possible due to adaptation and amplification of GBS in new niches but the evolutionary and functional mechanisms underpinning those phenomena are poorly known. Based on analysis of 1,254 curated genomes from all major GBS host species and six continents, we found that the global GBS population comprises host-generalist, host-adapted and host-restricted sublineages, which are found across host groups, preferentially within one host group, or exclusively within one host group, respectively, and show distinct levels of recombination. Strikingly, the association of GBS genomes with the three major host groups (humans, cattle, fish) is driven by a single accessory gene cluster per host, regardless of sublineage or the breadth of host spectrum. Moreover, those gene clusters are shared with other streptococcal species occupying the same niche and are functionally relevant for host tropism. Our findings demonstrate (1) the heterogeneity of genome plasticity within a bacterial species of public health importance, enabling the identification of high-risk clones; (2) the contribution of inter-species gene transmission to the evolution of GBS; and (3) the importance of considering the role of animal hosts, and the accessory gene pool associated with their microbiota, in the evolution of multi-host bacterial pathogens. Collectively, these phenomena may explain the adaptation and clonal expansion of

(NCBI), with accession numbers provided in S1 Table. Phylogenetic analysis of the 1,254 Group B Streptococcus genomes, together with all metadata, is available at the project URL https://microreact.org/project/vkEcchaHsSJa6sLhSGvmHF-group-b-streptococcus-host-adaptation-2023 within Microreact.

**Funding:** RNZ and MF received funding from a Technology Strategy Board, Agri-Tech Catalyst - Early Stage Feasibility grant (Ref 132168), for funding the mutagenesis and *in vivo* studies. SDB received funding from the Bill & Melinda Gates Foundation (Opportunity INV-010426) for the JUNO Project, which funded sequencing of part of the isolates included in this study. DJ and in part CC salaries were provided by the JUNO project. The funders had no role in study design, data collection and analysis, decision to publish, or preparation of the manuscript.

**Competing interests:** A patent for Group B *Streptococcus* (GBS) antigens associated with strains virulent in fish has been filed by the Moredun Research Institute. MF and RZ are named inventors on this application. The International Patent Application number is WO 2016/034879 Al. This application covers GBS genes required for virulence in fish, i.e. Locus 3 as described in this manuscript.

GBS in animal reservoirs and the risk of spill-over and spill-back between animals and humans.

## Author summary

Group B *Streptococcus* (GBS) is a bacterium that represents a health and food security threat in three major host groups: humans (in particular neonates), bovines, and fishes. However, the genomic mechanisms driving adaptation to these hosts remain unclear. Here, we use powerful statistical approaches to compare genomes of GBS from around the world. We found that GBS' ability to exchange genes is a good indicator of how well it can adapt to different hosts. Additionally, three groups of genes appear crucial for GBS to infect either humans, cows, or fishes, regardless of the bacterial strain. These gene groups are also found in other similar bacteria that live in the same hosts or environments. Where a functional role is already known for two gene groups in humans and bovines, respectively, it is shown here for the first time for the third gene group in fishes. Our findings demonstrate the importance of considering the role of animals in the evolution of multi-host bacterial pathogens like GBS. Gene exchange between bacteria infecting multiple hosts represents a high threat to human health, as high-risk types of GBS could adapt and expand in animals, which could then act as reservoirs for highly pathogenic human infections.

## 1 Introduction

Group B streptococci (GBS) were first described by Rebecca Lancefield, who isolated them from "certified milk" and milk of a cow with mastitis [1], giving the only member of GBS its scientific name, *Streptococcus agalactiae*. Since then, GBS has become a well-recognized human pathogen, primarily affecting neonates and their mothers [2, 3]. It is also increasingly a cause of disease in non-pregnant adults, both as opportunistic infection in immunocompromised individuals [4] and as foodborne disease in adults without co-morbidities [5–7]. The public health burden posed by GBS, particularly on maternal and child health in low—and middle-income countries, is such that the World Health Organisation has identified the development of GBS vaccines for maternal immunization as a priority [8, 9]. Like group A streptococci (GAS, *Streptococcus pyogenes*) [10], GBS not only causes a spectrum of diseases but can also be carried asymptomatically. In contrast to GAS, GBS is primarily carried in the gastrointestinal and urogenital tract and, possibly more important from an evolutionary perspective, it also has a wide range of animal hosts.

GBS continues to be an important cause of bovine mastitis around the world, with negative impacts on milk quality and quantity, cow health, and farmers' livelihoods [11]. It was largely eliminated from northern Europe, but re-emerged in the 21st century [12–14]. GBS is also a major pathogen of tilapia, the world's third most farmed fish species (FAO, 2022). It was first described in poikilothermic species (fishes, frogs) in the 1980s [15] and subsequently emerged as a major fish pathogen during the global expansion and intensification of fish farming [7, 16].

Multi-host bacterial pathogens can adapt to host species using diverse mechanisms. These range from point mutations [17] to the acquisition—through horizontal gene transfer (HGT) —of accessory genome content that provides a survival advantage in the context of a particular host, e.g., the ability to evade the host immune system or the acquisition of new metabolic

pathways in response to the availability of particular nutrients [18, 19]. For example, a transposon carrying *scpB*, encoding a C5a-peptidase (a surface-associated serine protease) [20], which enables invasion of epithelial cells, has been associated with human GBS [21, 22], whereas nutritional adaptation is exemplified by the acquisition of a mobile genetic element (MGE) encoding a lactose-fermenting pathway, which appears to be critical for successful colonisation of the bovine mammary gland [23]. Considering the high genome plasticity of GBS [24], and the emergence of new clades in animal host species in the last hundred years [7, 13, 25], there is a need to gain a better understanding of mechanisms driving host adaptation, including HGT, not least to ensure that the emergence of new variants can be monitored during the anticipated introduction of human GBS vaccination (WHO, 2021).

Here, we analyse the GBS population using 1,254 genomes, rigorously selected to represent diversity in lineages, host species, and geographic origins, to gain insight into the role of genome plasticity and the accessory genome in GBS host adaptation. To facilitate this analysis, we propose nomenclature based on clonal groups (CG) and sublineages (SL), as recently introduced for other multi-host pathogens, and demonstrate that they comprise host generalists and host specialists lineages, whereby the latter can have a preferred or dominant host species (hereafter termed "host adapted") or be limited to one host species or host group (hereafter termed "host restricted"). Using genome-wide association studies (GWAS), we show that human-, bovine-, and fish-adaptation of GBS are associated with C5a-peptidase (*scpB*), the lactose operon (Lac.2) and an accessory gene cluster known as Locus 3, respectively, regardless of GBS lineage, and that those three gene clusters seem to be largely necessary and sufficient to explain host adaptation. Through *in vivo* challenge experiments, we provide functional evidence for the critical role of Locus 3 in fish-associated GBS. This has important repercussions on public health, such as the adaptation and clonal expansion of the hypervirulent clone ST283 in fish, which was integral to the unexpected GBS foodborne-outbreak in Singapore in 2015, and to the current widespread dissemination of this clone in South-East Asia [7, 26, 27].

## 2 Materials and methods

### 2.1 Ethics statement

Animal experiments were in compliance with the ARRIVE guidelines and were carried out out in accordance with the U.K. Animals (Scientific Procedures) Act, 1986 and associated guidelines, and with the EU Directive 2010/63/EU for animal experiments. Challenge studies were conducted by a contract research organisation specialized in aquatic animal health (Ictyopharma, France). Studies were performed to Good Laboratory Practice standards, approved by an internal Ethics Committee and the French Ministry of Research, and audited by an independent Quality Assurance partner.

### 2.2 Genome selection

The aim of this study was to explore genomic mechanisms of host adaptation in GBS at the population level, and in particular across its sublineages. To this end, we applied in-depth comparative genomic approaches, such as GWAS, to a comprehensive global GBS dataset with a well-balanced proportion of human and animal isolates, and therefore a low bias towards human GBS. A detailed description of all materials and methods can be found in Section A in S1 Appendix.

First, we aimed to collate a dataset representative of the broad diversity of the GBS population in terms of clonal complexes, hosts of origin, geographical locations, and temporal range (n = 2,437) (S1 Table). Metadata were curated from the literature (March 2020) and whole genome sequence data were obtained either from public repositories or self-generated. Using

Pandas v1.1.3 [28], we de-duplicated the initial dataset for clones that were over-represented based on a series of metadata variables and further filtered based on genome assembly quality (see section A.2 of the S1 Appendix). The resulting de-duplicated, quality-filtered dataset comprised 1,254 genomes.

## 2.3 Core genome analyses

IQ-TREE v.2.0.6 [29] was used to estimate a core genome phylogeny with a GTR model, from a recombination-free alignment file obtained with snippy v4.4.5 (https://github.com/tseemann/snippy) and gubbins v3.2.0 [30], using NGBS128 as reference. On the same alignment file, fastbaps v1.0.8 [31] was used to define genomic clusters.

A generalised linear model was run in RStudio v2022.07.01, R v4.2.0 (2022–04-22), on the output from gubbins, after having mapped the internal nodes and the leaves to the corresponding CG, to test association between the host-specialisation level (generalists, adapted, restricted) and the number of nucleotide bases identified in recombination blocks.

## 2.4 Accessory genome analyses

Accessory gene distances were calculated with GraPPLE (https://github.com/JDHarlingLee/GraPPLE) using the Jaccard similarity index from a gene presence/absence matrix; the latter was generated with panaroo v1.2.0 [32] from gff files annotated with Prokka v1.14.5 [33]. The resulting file was visualised with Graphia v2.2 [34].

## 2.5 Genome-wide association studies

Scoary v1.6.16 [35] was run to detect gene-enrichment in GBS from the three major host groups from the panaroo-generated presence/absence gene matrix. The host groups (human, bovine, and the poikilotherm group including fishes and frogs) were defined as binary phenotypes, with 1 as belonging to the host, and 0 as not belonging to the host; GBS genomes from host groups other than these three were always categorised as 0, including those originating from dead fish sampled at markets for which a possible contamination due to human handling could not be excluded.

The pyseer suite v1.3.3 [36] was used to assess unitig association with a linear mixed model on the same host groups as above.

## 2.6 *In vivo* assessment of the role of Locus 3 in fish infection

Challenge studies on Nile tilapia (*Oreochromis niloticus*) were carried out with ST7 and ST283 knock-out mutants of Locus 3 (ΔLocus3). Strains, plasmids and oligonucleotide primers used for mutagenesis are described in S1 Appendix (Table C.1 and Table C.2). In preparation for challenge, tilapia were transferred into 100 L experimental tanks (four replicates per GBS strain, two negative control replicates) under the same ecological conditions. For challenge experiments, ten fish from each experimental tank were challenged by intraperitoneal (IP) injection with 0.1 mL of phosphate buffered saline (PBS; negative control) or 0.1 mL of $10^5$ CFU/mL of GBS in PBS. For contact challenge experiments, IP-challenged tilapia were co-habited with 40 healthy tilapia per tank.

Mortality of IP-challenged and contact-challenged fish was monitored for 21 days. Moribund fish were euthanized and included in mortality counts. Kidney and brain samples were taken from up to 10 fish per tank and used to check for presence of GBS (bacterial culture) and Locus 3 genes (PCR assays).

Kaplan-Meier survival analysis [37] was conducted to compare survival of knock-out mutants and their isogenic wild type strains. A statistical Log-rank test was conducted [38], using Microsoft Excel software, to compare survival curves of tilapia challenged by wild type vs knock-out mutants.

### 2.7 Time-scaled phylogenies of two clones of public health interest: CG283 and SL23

BEAST v2.6.6 [39] was used to estimate the time of emergence of two clones of public health interest, CG283 and SL23. The best model parameters chosen were: GTR+G4 with a strict clock and a constant coalescent population size. Xml files were run in triplicate with 200 million generations and sampling frequency of 10,000 (burn-in 10%). Resulting trees from the three replicate runs were combined with LogCombiner, and the final tree files were obtained with TreeAnnotator and visualised with FigTree v1.4.4 (http://tree.bio.ed.ac.uk/software/figtree/) and Microreact [40].

The Maximum Clade Credibility trees from the two BEAST datasets were used together with gene presence/absence for *scpB*, Lac.2, and Locus 3 in a discrete traits analyses, as well as with host. To infer the gene presence/absences at the ancestral nodes in the trees, the ancestral character estimation function (ace) within R-package ape [41] was used with a discrete asymmetric model (all rates different, ARD).

## 3 Results and discussion

### 3.1 The global GBS population is composed of host-generalist and host-specialist lineages

The GBS global population comprises: i) host generalists, defined here as having no more than 80% of isolates originating from a single host species (human, bovine, camel) or host group (poikilothermic species, including fishes and frogs; for simplicity, this host group is referred to as "fish" throughout the manuscript), and host-specialist lineages, defined here as having more than 80% of isolates originating from a single host or host group, and further subdivided into ii) host-adapted lineages, which have a clear host predilection, defined here as having more than 80% but less than 98% of isolates originating from a single host or host group; and iii) host-restricted lineages, whose occurrence is almost exclusively associated (≥98%) or completely restricted to a single host species or host group (Fig 1A). The cut-offs were chosen by plotting in a histogram the prevalence of the dominant host within each CG (e.g., CG23: dominant host bovine, prevalence 64.78%; CG22: dominant host human, prevalence 100%) (Fig 1B) (see Section A.3 in S1 Appendix).

Of 15 sub-populations that were identified by hierarchical Bayesian clustering, largely in alignment with the topology of the core gene phylogeny (Fig 2A), five were host generalists, i.e. sublineage (SL) 1, SL23, SL103, SL283, and SL314; four were host-adapted, i.e. SL17, SL19, SL26 and SL130 (all with a predilection for the human host); and the remaining six were host restricted, i.e. SL22 (human), SL61, SL91 (bovine), SL552 (fish), SL609 and SL612 (camel). In this nomenclature, sublineages are named after the most common sequence type (ST) in each subpopulation based on 7-gene multi-locus sequence typing (MLST) nomenclature (Fig C.1 in S1 Appendix). Within each SL, clonal groups (CG, n = 23) were defined based on phylogenetic sub-clusters using the same nomenclature principle as for SL (Fig C.1 in S1 Appendix). Among CG, the same cut-offs used for SL identified nine host generalist CG, six host-adapted CG, and eight host-restricted CG (Figs 1 and 2B). Two generalist SL (SL1, SL283) comprised three generalist CG each (CG1, CG459, and CG817; CG6, CG7, and CG283, respectively), and

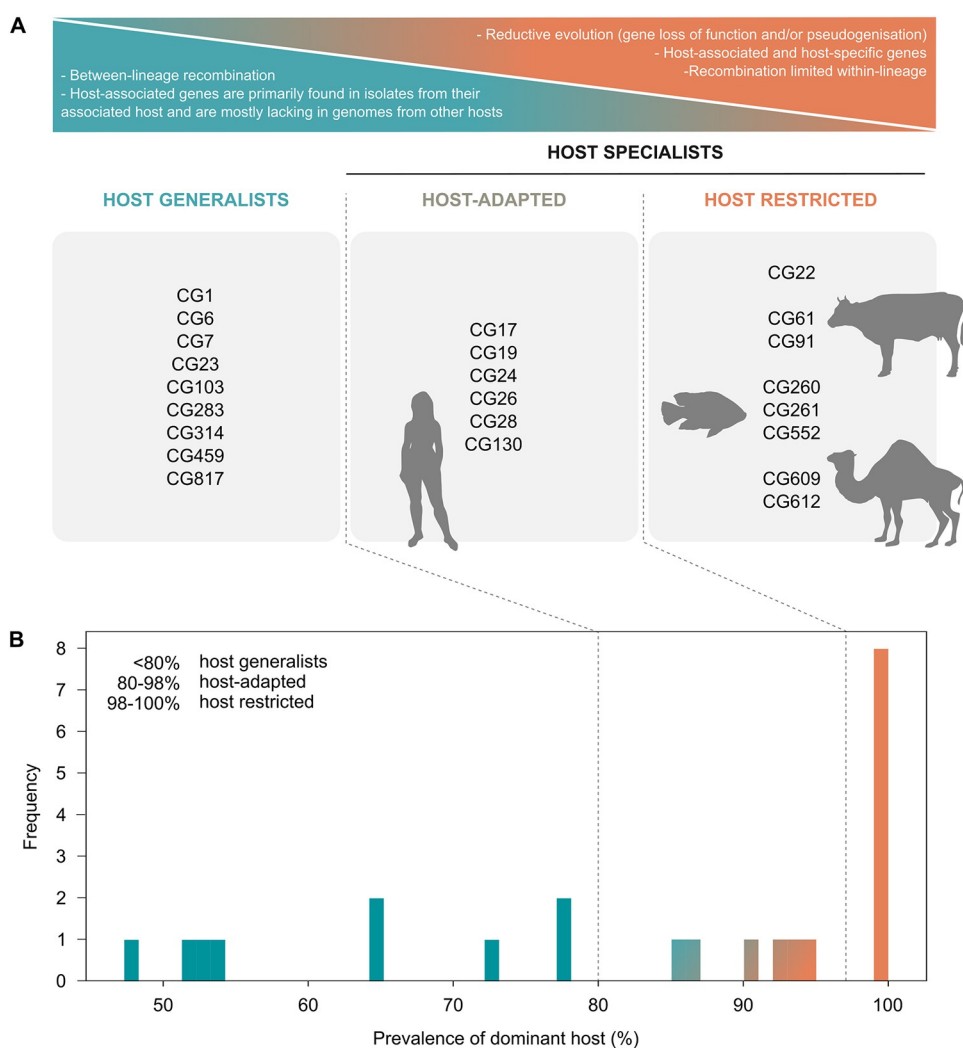

**Fig 1. Diagram illustrating host specialism levels in Group B *Streptococcus* (GBS) clonal groups (CG).** A) Host generalist lineages show extensive between-lineage homologous recombination and the three host-associated accessory gene clusters (*scpB*, Lac.2, Locus 3) are primarily found in isolates from their associated host, while they are lacking from isolates from other hosts. Host restricted lineages can be associated with reductive evolution (e.g., gene loss of function and genome reduction as in CG260, CG261 and CG552), they carry host-associated genes (e.g., CG260, CG261 and CG552 all carry Locus 3) and they either show absence of recombination (CG260, CG261 and CG552) or recombination limited within CG (e.g., CG61 and CG91). Host-adapted CG are primarily associated with one host and they show all the characteristics of the host restricted lineages except for genome reduction and pseudogenisation. B) Thresholds used for the categorisation of CG in the three levels of host specialism; the prevalence of genomes within each CG associated with the dominant host species (x-axis) was plotted to identify cut-offs, where the y-axis represents the number of CG corresponding to a given host prevalence.

one host restricted SL comprised three restricted CG (SL552, including CG260, CG261, and CG552). However, SL23, which was classified as host generalist, encompassed the host generalist CG23 as well as the host-adapted CG24, which is primarily found in humans. The two CG are associated with different capsular types: the human-adapted CG24 primarily comprises isolates from capsular type Ia, whereas CG23 is mostly associated with capsular type III, except for a human-associated subclade of type Ia and II isolates (for more detail, please refer to our Microreact project at microreact.org).

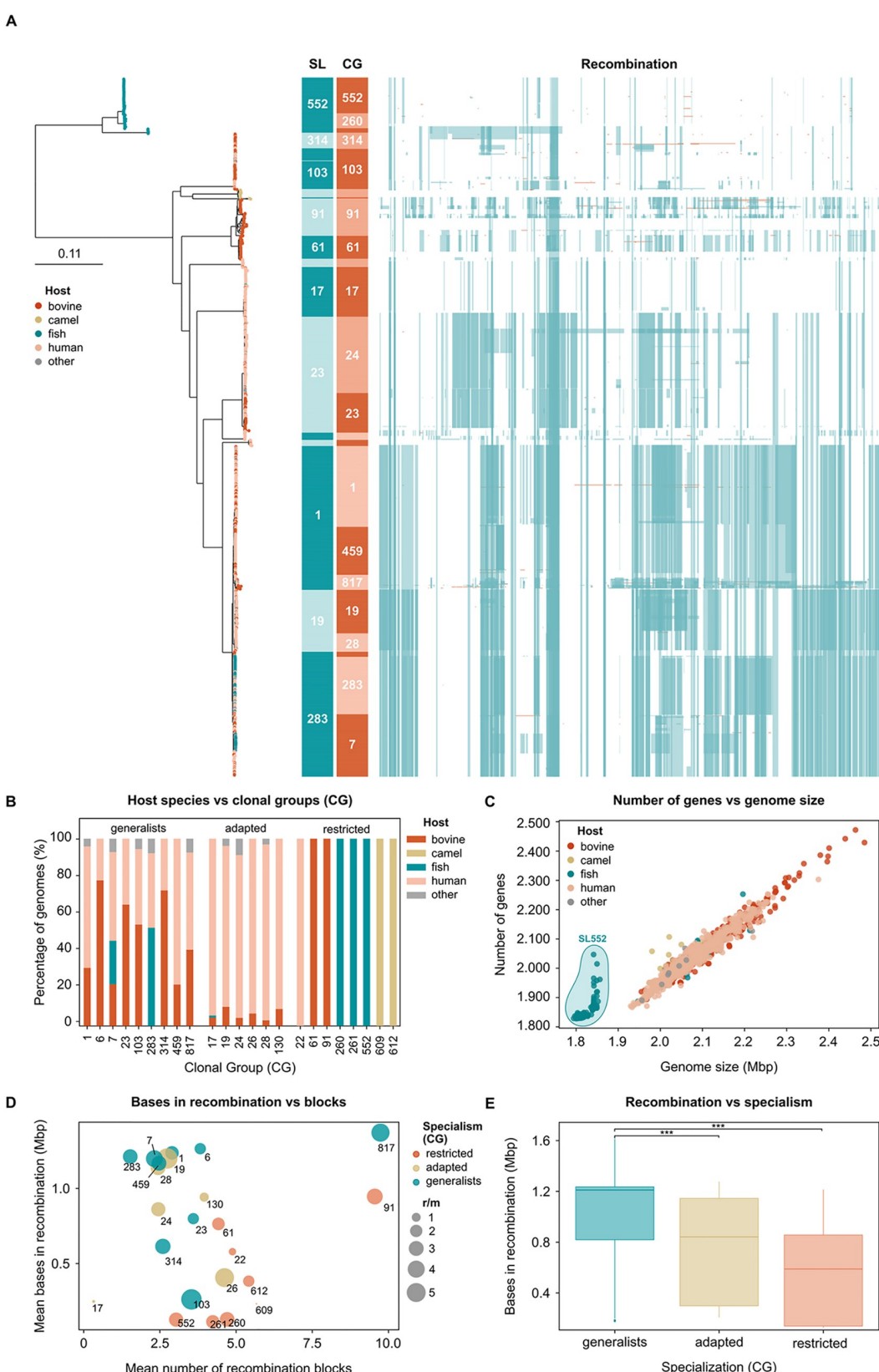

**Fig 2. Population structure and pangenome of Group B *Streptococcus* (GBS).** A) Maximum-likelihood phylogenetic tree of 1,254 GBS genomes; leaf colours indicate host of origin, and external strips show sublineage (SL), clonal group (CG) and homologous recombination; tree was rooted on an out-group of five reference genomes from *Streptococcus pyogenes* (hidden); B) Prevalence of host species within each CG; C) Correlation of number of genes and assembly size; SL552 shows a marked pseudogenisation and reduced genome size; D) Correlation of average number of recombination bases and recombination blocks of each CG, as well as recombination to mutation (r/m) rate; E) Recombination observed in the three categories of host-specialism; the difference between groups is statistically significant (p-value<0.0001).

The existence of host-generalist and host-specialist lineages within populations of multi-host bacterial pathogens is well-described, for example in *Salmonella enterica* [42], *Campylobacter jejuni* [43, 44], and *Staphylococcus aureus* [18, 19]. These observations have been associated with genomic phenomena that may lead to host-restriction, typically in the case of pseudogenisation and/or gene loss, as described for SL61 [45] and SL552 (Fig 2C) [46], or to host jumps, which provide access to a new accessory gene pool, with subsequent host-adaptation, which is often a result of acquisition of accessory genes that provide an adaptive advantage to the new ecological niche.

### 3.2 Host-specialism among GBS lineages is associated with different levels of genome plasticity

Recombination can also play a role in host-adaptation [47]. We observed marked differences in recombination between clonal groups of GBS, ranging from large recombination blocks in, e.g., host generalists CG283 and CG817 to absence of recombination in, e.g., host-restricted CG552, CG260, CG261 (Fig 2A and 2D). Host-generalist lineages appear more subject to recombination than host-specialists, as illustrated by the number of nucleotide bases present within recombination blocks, for which the difference is statistically significant (t-value generalists = 107.56, adapted = -15.42, restricted = -28.12, p-value<0.0001) (Fig 2E).

Our results are indicative of a higher genome plasticity of host-generalist lineages compared to host-specialists, in particular to host restricted lineages, and suggest that the ability to uptake and retain foreign DNA, including accessory genes that could provide a survival advantage in new niches and hosts, is superior in host-generalists (e.g., SL283). The lack of shared recombination between some lineages (e.g., SL1, SL19 and SL283 vs others; SL23 vs others; SL91 vs others, SL61 vs others), coupled with results from analysis of the accessory gene set (section 3.3), suggests the existence of some barriers to genetic exchange (i.e., HGT) within GBS. These barriers are unlikely to be ecological (e.g., segregation [48]), as at least some lineages co-exist in the different host populations and in the same geographical areas. As an example, GBS isolates belonging to ST1 (part of SL1/CG1) were reported in both humans and cattle in Colombia [49] and in several Northern European countries [50, 51], where ST23 (part of the SL23) was also detected in both hosts. Rather, these are more likely mechanistic barriers that can act as a defense against the uptake of foreign DNA, such as restriction-modification systems (RMS), CRISPR, or antiphage systems, as described by Mourkas and colleagues [48].

### 3.3 The accessory genome in GBS is lineage-associated

To explore the role of the accessory genome content in host adaptation, we used network analysis of accessory gene distances. This shows distinct clusters of accessory genome content that align with SL (and therefore with CG) rather than host species (Fig 3), noting that SL and host species are inherently fully concordant for host-restricted lineages, such as SL552 or SL91. These observations suggest that the accessory gene set as an ensemble depends more on the sublineage of origin of the isolates than on their ecological niche in GBS. This is in contrast to

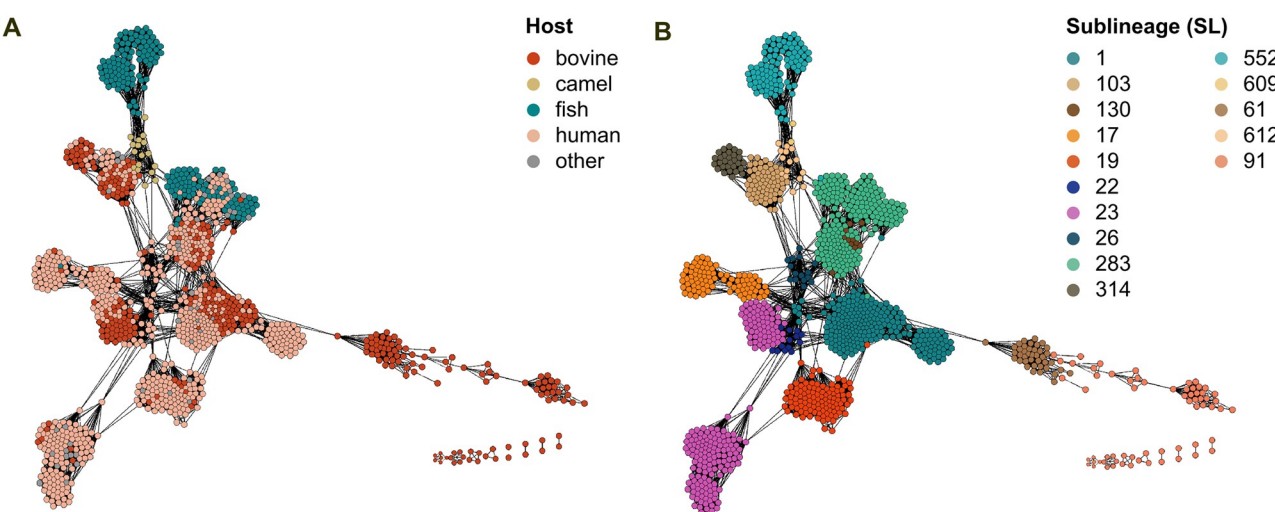

**Fig 3. Accessory gene distance network of 1,254 Group B *Streptococcus* (GBS) genomes.** A) Major host groups (human, bovine, fish and camel) are overlaid to the nodes; B) Sublineages (SL) defined by fastbaps and renamed based on an inheritance principle from 7-gene MLST nomenclature are shown. The two panels show association of accessory gene clusters with SL, and a lack of clustering based on host species, unless when this is a direct result of host-specific lineages (e.g., SL552, SL91, SL61).

what is reported in *S. aureus* by Richardson and colleagues [19], who showed an association between host species and the whole set of accessory genes. These results, coupled with previous knowledge on the association of MGE encoding host-associated virulence factors or metabolic pathways (e.g., *scpB* in humans [20, 52] and Lac.2 in bovines [23, 51, 53]), led us to hypothesise that a limited number of acquired genes might drive host-adaptation in GBS.

### 3.4 Specific gene clusters drive host-adaptation in GBS

The hypothesis that a specific subset of genes drives host association was tested with GWAS using two approaches, one pangenome-based and the other unitig-based. Both analyses identified three accessory gene clusters which were strongly associated with humans or animal hosts, as detailed below.

**3.4.1 GBS in humans: The *scpB* transposon.** The pangenome-based approach identified the *scpB* transposon as significantly positively associated with the human host (*p*-value for *scpB* was $1.26 \times 10^{-131}$; all *p*-values reported in this paper for pan-GWAS analyses were corrected with the Benjamini-Hochberg method) (S2 Table), and similar results were obtained with the unitig-based approach (Fig 4A and C.2A in S1 Appendix). The *scpB* gene is a transposon-encoded virulence factor that is known for its higher prevalence among human isolates [21, 52]. It has been shown both through *in vitro* and *in vivo* approaches how this gene interacts negatively with the human host immune system (cleaving the C5a complement component) [54, 55] and how it contributes to GBS cellular adhesion and invasion by binding to fibronectin [56, 57]. Additionally, it has been associated with the ability to colonise or infect the human host in other human pathogenic streptococci (Group A *Streptococcus*, *Streptococcus dysgalactiae* subsp. *equisimilis* and *Streptococcus canis*) [20, 52]. *In vitro* work suggests that *scpB* likely plays no role in bovine GBS infections [20], even when it is carried by GBS cattle isolates. Its role in fish infections has not been assessed. Interestingly, in our dataset, *scpB* in GBS from fish was uniquely found in a lineage shared with humans (SL283),

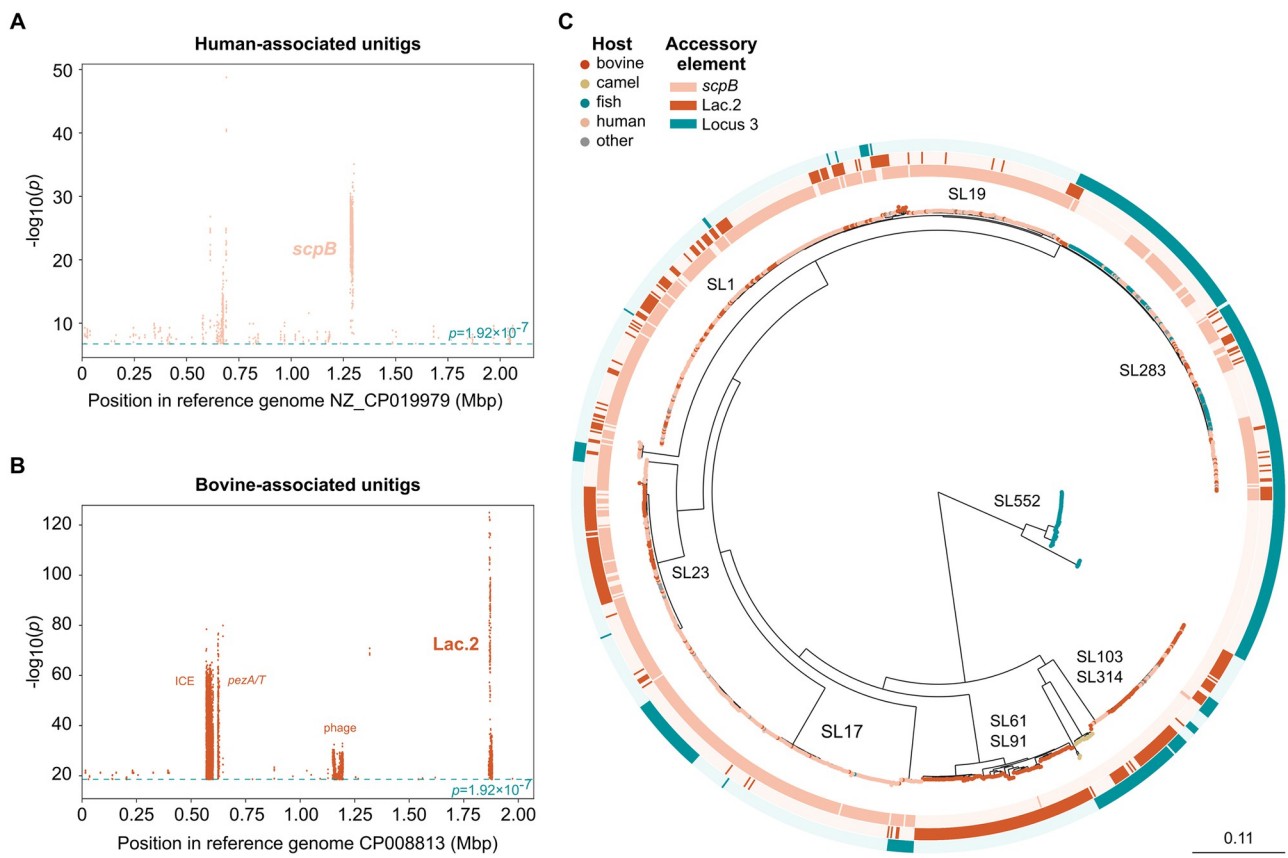

**Fig 4. Host-associated accessory gene clusters in Group B *Streptococcus* (GBS).** A) Manhattan plot of human-associated unitigs mapped to reference genome NZ_CP019979 (only genes/elements detected as significant by both GWAS methods are here annotated); B) Manhattan plot of bovine-associated unitigs mapped to reference genome CP008813; C) Circular maximum-likelihood phylogenetic tree of 1,254 GBS genomes. Leaf colours indicate host of origin, whereas the three external strips show presence/absence of the three main host-associated accessory gene clusters (human-*scpB*, bovine-Lac.2, fish-Locus 3, respectively).

which could explain why this is the only lineage from fish that shows evidence for zoonotic transmission (see section 3.6.1).

**3.4.2 GBS in cattle: The lactose operon Lac.2.** In bovine GBS, significant genes with both GWAS approaches corresponded to a 9 to 11-gene cluster, the Lac.2 operon, in particular its genes *lacEG* (Scoary *p*-value $4.69 \times 10^{-168}$; Fig 4B and C.2B in S1 Appendix, and S3 Table). In GBS, a Lac.2-positive genetic background is associated with phenotypic lactose fermentation, which has been observed primarily in cattle GBS isolates with *in vitro* and transcriptomic approaches [51, 53]. In cattle, GBS only causes infection of the mammary gland, which is rich in lactose (milk sugar). The acquisition of metabolic pathways (here: Lac.2) in response to nutrient availability (here: lactose) is a known driver of niche adaptation [43, 47, 58], and fermentation of lactose via lactose operons is known to promote growth in lactose-rich environments such as the bovine udder not only in GBS, but in other mastitis-causing Gram-positive (*Streptococcus uberis*, *Streptococcus dysgalactiae* subsp. *dysgalactiae* [23, 53]) and Gram-negative organisms (*Klebsiella pneumoniae* [58]), which strengthens the argument that acquisition of lactose-fermenting genes drives niche adaptation to the bovine mammary gland.

**3.4.3 GBS in fish: Locus 3.** In fish GBS, the unitig-based approach was unsatisfactory, which was likely due to a strong population structure effect, as genomes from this host group

are found only in two SL (SL552 and SL283). With the pangenome-based approach, genomes from fish GBS appeared to be enriched for genes belonging to a cluster known as Locus 3 (SE 99.5%; SP 72.6%; BH *p*-value 9.62x10$^{-85}$) (S4 Table). Locus 3, which is described as fish-associated [59], was present in 99.5% of fish and frog GBS genomes and only in a minority of non-fish assemblies, especially in the generalist SL283, a SL primarily shared between humans and fish. Unlike for *scpB* in humans and Lac.2 in bovines, there was no prior evidence for a functional role of Locus 3 in fish (see section 3.5). However, we detected a conserved segment of this element in all available genomes (n = 212, NCBI, June 2024) from another important agent of fish streptococcosis, *Streptococcus iniae* (Fig C.3 in S1 Appendix) [60], highlighting its association with the aquatic niche and its possible impact in streptococcal adaptation to fish.

The significant association of the three accessory gene clusters with the three main GBS host groups across SL and CG (Fig 4C) as well as with other streptococci pathogenic to the same host species, suggests that they are each major drivers of adaptation to the distinct host niches in GBS lineages, and possibly more broadly in the genus *Streptococcus*. As mentioned above, functional evidence for the role of *scpB* in human infections and of Lac.2 in bovine mastitis already exists (sections 3.4.1 and 3.4.2). This was not the case for Locus 3, which is why we decided to proceed with *in vivo* challenge experiments in fish after mutagenesis experiments.

## 3.5 Functional evidence of the role of locus 3 in piscine GBS

Locus 3 is a 17-gene cluster inserted between a GNAT family N-acetyltransferase and a class I SAM-dependent methyltransferase, which includes, among others, genes for carbohydrate transport and metabolism [59] (Fig 5A). To test the functional relevance of this genomic determinant of fish-association of GBS, Nile tilapia (*Oreochromis niloticus*) were challenged with a multi-gene knock-out mutant (ΔLocus3) and wildtype (WT) isolates of two highly virulent fish-associated GBS strains (ST7 and ST283). Animals were either challenged intraperitoneally (IP) or through cohabitation with the IP-challenged fish. For both challenge routes and both GBS strains, a significant reduction in mortality was observed in groups challenged with the ΔLocus3 mutant relative to WT (Fig 5B–5E and Table C.3 in S1 Appendix). The observed attenuation of GBS after removal of Locus 3 provides the first functional evidence of its importance in GBS infection of fish.

## 3.6 The impact of host-associated gene clusters on public health

**3.6.1 CG283, a generalist clade associated with foodborne infection.** CG283 (Fig 6A) is widespread in fishes in the South-East Asian region [7] and it is carried asymptomatically by close to 3% of the human population in some areas [61]. In 2015, it was recognized as the causative agent of the first ever reported outbreak of GBS foodborne disease. All CG283 isolates in our study carried the fish-associated Locus 3, regardless of whether they originated from fish or humans, which is probably due to the fact that most human ST283 originated from people infected through fish consumption [26]. Only a subset of ST283 isolates carries *scpB*, as previously observed in Vietnam and Thailand [62]. Traits analysis predicted *scpB* to be present in the common ancestor of CG283 isolates, and to have been lost multiple times, with two major events corresponding to two branches (the first around 1992, and the second around 2003) (Fig 6A). The presence of *scpB* in the ancestral node of CG283 suggests a probable human origin for this clade, which would have acquired Locus 3 concomitantly to the expansion of aquaculture (1980s), illustrating how initial spill-over from humans to animals can be followed by host-adaptation and subsequent amplification in the new host, a phenomenon also described in, e.g., *S. aureus* [18]. We hypothesise that the exchange of accessory genetic material between human and fish GBS is favoured by frequent direct or indirect contact between humans and

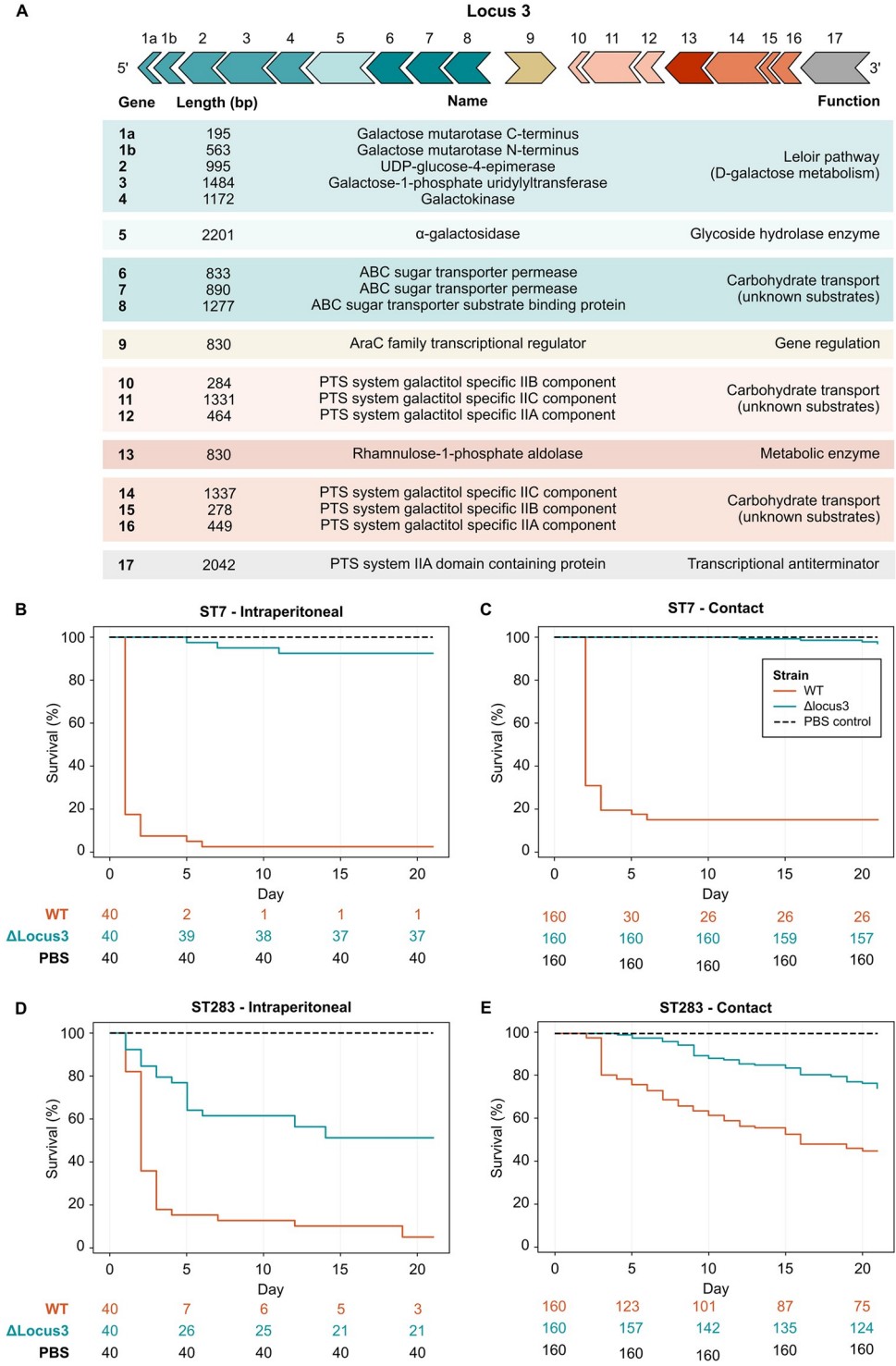

**Fig 5. Locus 3 gene cluster and survival curves of Nile tilapia (*Oreochromis niloticus*).** A) Locus 3 gene cluster organisation and associated gene functions; B) Tilapia challenged intraperitoneally (IP) with Group B *Streptococcus* (GBS) Sequence Type (ST) 7 wild type (WT) or its isogenic mutant (ΔLocus3); C) Tilapia challenged with GBS ST7 (WT) or its isogenic mutant (ΔLocus3) through cohabitation with IP-challenged fish; D) Tilapia challenged IP with GBS ST283 (WT) or its isogenic mutant (ΔLocus3); E) Tilapia challenged with GBS ST283 (WT) or its isogenic mutant (ΔLocus3) through cohabitation with IP-challenged fish. All experiments (B-E) included a negative control (mock challenge with phosphate buffered saline). Below survival curves, number at risk has been indicated for each timepoint.

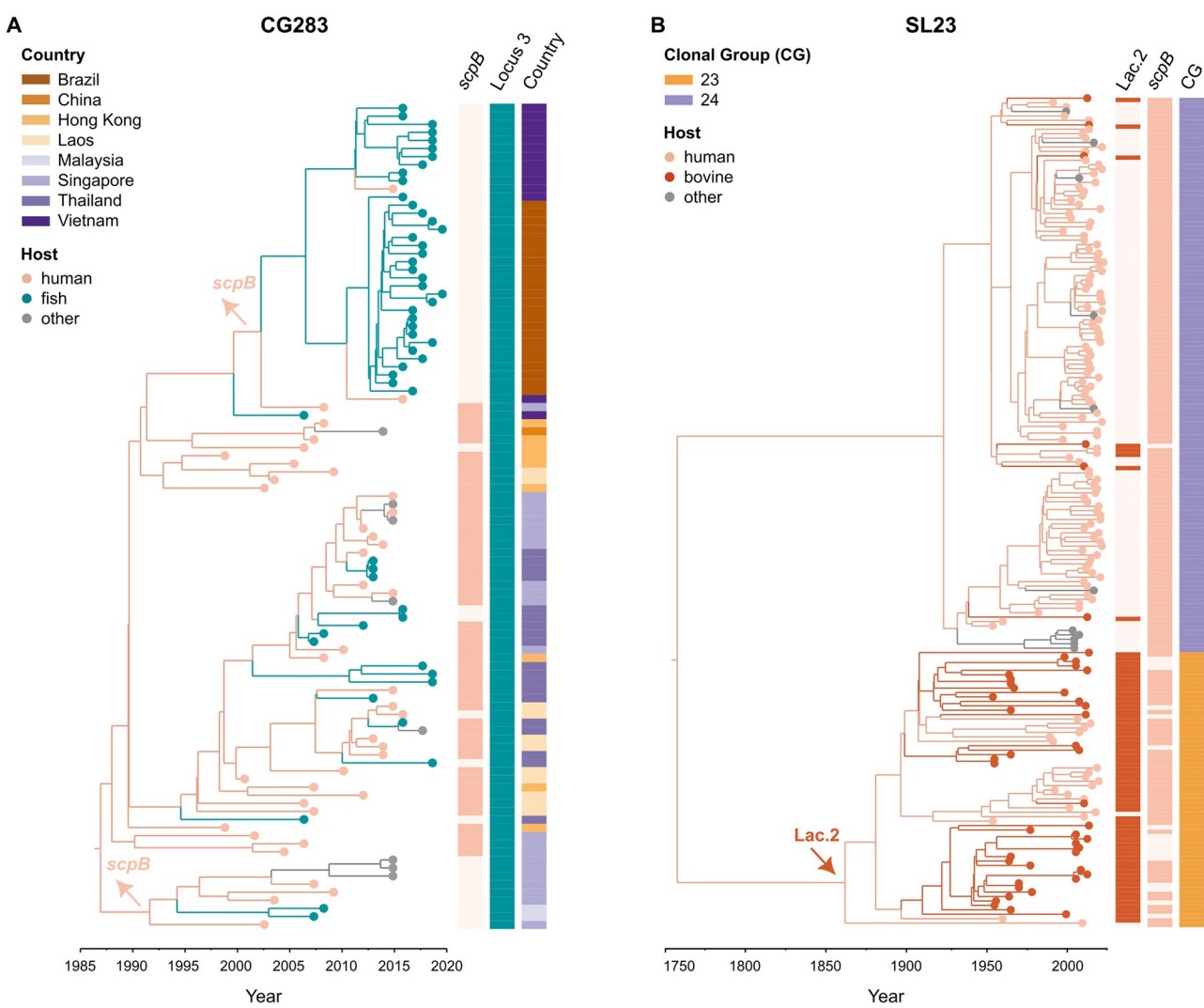

**Fig 6. Time-scaled phylogenies of two clones of public health interest in Group B *Streptococcus* (GBS).** Branch colour shows host-jumps, leaf colour the host of isolation, and external strips show presence-absence of genes and features of interest. A) Clonal group (CG) 283 is predicted to have originated around 1986 from humans. Its population shows diversification in multiple sub-clades over the years. Genomes associated with human cases from the 2015 outbreak of foodborne infection in Singapore are part of a sub-clade in which most of the genomes carry the human-associated transposon bearing *scpB*. In recent years, CG283 has been described in Brazil (first isolates available are from 2015); this introduction likely happened from Vietnam, and it is associated with a *scpB*-negative genotype. The two main *scpB* loss events are indicated with arrows on the phylogeny; B) Sublineage (SL) 23 likely originated in humans at the beginning of the 18th century. It is described as a host-generalist clonal complex (CC). However, when assessing host prevalence within its two CG (CG23 and CG24), as well as the distribution of human and bovine-associated accessory gene clusters (*scpB* and Lac.2, respectively) the two CG show distinct patterns of host-tropism. CG24 appears as human-adapted, whereas isolates of CG23, whose common ancestor is predicted to have acquired Lac.2 in the second half of the 1800s (red arrow), show a tendency towards bovine-specialism, but retain the ability to cause human infections (*scpB* is highly conserved).

fish in South-East Asia, e.g., through recycling of human waste as fish feed, floating fish farms or fish handling and consumption. A CG283 sub-clade mostly carrying *scpB* (Fig 6A) is associated with human infections, and it includes genomes from GBS isolated during the 2015 Singaporean foodborne outbreak that was linked with the consumption of contaminated raw fish [7, 26], an outbreak that resulted from spill-back of the now fish-adapted CG283 into humans. In this sub-clade, however, *scpB* appears to have been lost on multiple occasions. The second major loss of *scpB*, which happened around 2003 (Fig 6A), appears to have taken place in

Vietnam, and was followed by expansion of a CG283 sub-clade primarily associated with fish. The emergence of CG283 in Brazil, estimated around 2012 but whose first available isolates date to 2015, likely resulted from an introduction of the Vietnamese sub-clade, as exemplified by the Brazilian isolates clustering within the Vietnamese ones (Fig 6A). So far, very few human infections with GBS ST283 have been reported from Vietnam [7], and no human infections associated with CG283 have been reported in South America. This is probably not a result of differences in eating habits compared to the Asian population, as a percentage as high as 43.45% of the Brazilian population is reported to consume raw or under-cooked seafood [63]. Rather, absence of reported CG283 infections in humans could be due to the absence of *scpB*. The evolution of CG283 illustrates how ongoing surveillance of GBS across host species is needed to detect emergence of new clades as well as the acquisition and loss of accessory genome content that informs on the hazard to public health and food security.

**3.6.2 SL23, a historical host specialist that gave rise to a host generalist.** SL23, which was referred to as CC23 in previous literature, is an old GBS sublineage, with emergence estimated to have occurred in the middle of the 18[th] century in our study and by others [25]. It has long been considered a host-generalist, but there are two CG within SL23, with distinct patterns in terms of host tropism. The first one, CG24, is a host-adapted CG that primarily causes disease in humans (Fig 2), with occasional host jumps to cattle. These host jumps are associated with sporadic acquisitions of Lac.2 (Fig 6B). Reflecting its human host association, *scpB* is strongly conserved in this CG, with the exception of one bovine isolate. The second CG in SL23 is CG23, which is a host-generalist clade that primarily affects humans and cattle, with additional reports in, e.g., companion animals, aquatic mammals and reptiles [64]. Mirroring the situation for *scpB* in CG24, Lac.2 is highly conserved in CG23, with the exception of one human isolate (Fig 6B), explaining its ability to infect the bovine mammary gland. As in CG283, *scpB* has been lost multiple times independently from CG23, suggesting that this virulence factor is lost easily in the absence of selective pressure from the human host. Considering that the phylogeny suggests bidirectional interspecies transmission of CG23 (Fig 6B) ongoing monitoring of CG23 infections in both host species would be needed to gauge risk of bovine GBS to public health.

## 4 Conclusions

Multi-host bacterial pathogens often comprise lineages that show distinct patterns of host-tropism, which can vary from host generalism, such as CC130 and CC398 in *S. aureus* [19], to host restriction, such as for the serovars Gallinarum and Pullorum of *S. enterica* subsp. *enterica* in poultry [42]. A multitude of interacting ecological (e.g., physical segregation), genomic (e.g., ability of a genome to uptake DNA, homologous and non-homologous recombinations, gene loss and pseudogenisation) and adaptive mechanisms can affect host-range [48]. Genome plasticity and HGT within GBS have been described as major driving forces in the evolution of GBS [24]. Here, we show that those forces may differ between lineages, with higher genome plasticity in host-generalists than in host specialists.

As in other bacterial pathogens [19, 48], adaptive mechanisms that confer a fitness advantage within the context of a particular niche play a crucial role in GBS. Despite the diversity of lineages associated with each major host category, and the diversity of host categories associated with many major lineages, we have identified that three key accessory gene clusters seem largely necessary and sufficient to explain host adaptation. Two of those clusters, i.e., *scpB* in human GBS [20] and Lac.2 in bovine GBS [23], were known to be genomic and functionally relevant markers of host association, and we provide functional evidence for the critical role of Locus 3 in fish infections in different genetic backgrounds of GBS. Remarkably, these three

elements are not only present in GBS but are also found in other streptococcal species that share ecological niches with GBS (e.g., *scpB* in the human or zoonotic pathogens Group A *Streptococcus*, *Streptococcus dysgalactiae* subsp. *equisimilis* and *Streptococcus canis*, Lac.2 in the bovine mastitis pathogens *S. dysgalactiae* subsp. *dysgalactiae* and *S. uberis*, and Locus 3 in the fish pathogen *S. iniae*), suggesting their broader importance within the genus *Streptococcus* in each of these hosts or niches.

Through analysis of two multihost clades (CG283 and SL23), we show the impact of acquisition or loss of these three elements on host-adaptation and expansion following host-jumps, and illustrate the public health relevance of such genomic mechanisms for GBS. We propose that, by focusing genomic surveillance on clones with high potential for host-jumps (i.e., those with high levels of recombination, such as SL283) and their acquisition of host-associated accessory gene clusters, strategies and interventions to reduce the risk of interspecies transmission could be improved. In addition, genomic surveillance of GBS across host species will be critical when human GBS vaccines are rolled out, to monitor HGT of vaccine targets such as the capsule, a phenomenon already observed in GBS [65, 66]. Selective pressure induced by vaccination could favour emergence of new serotypes, as also observed for *Streptococcus pneumonaie* [67], but with a need to consider both human and animal hosts to understand and monitor the evolution of GBS.

## Supporting information

Supplementary data files for this article include:
**S1 Appendix. Supplementary data file.** Detailed materials and methods, additional results, and supplementary figures and tables.
(PDF)

**S1 Table. Dataset and results file.** Dataset of genomes used for this study, with accompanying metadata, accession numbers, and main results of genomic analyses.
(XLSX)

**S2 Table. Scoary results.** Scoary results for human-enriched genes.
(XLSX)

**S3 Table. Scoary results.** Scoary results for bovine-enriched genes.
(XLSX)

**S4 Table. Scoary results.** Scoary results for fish-enriched genes.
(XLSX)

## Acknowledgments

We would like to thank John Lees for bioinformatic support for pyseer. We would also like to thank Marta Mansos-Lourenco, Carla Parada-Rodrigues and Sylvain Brisse for constructive feedback on the manuscript.

## Author Contributions

**Conceptualization:** Chiara Crestani, Michael Fontaine, Ruth N. Zadoks.

**Data curation:** Chiara Crestani.

**Formal analysis:** Chiara Crestani, Samantha J. Lycett.

**Funding acquisition:** Stephen D. Bentley, Michael Fontaine, Ruth N. Zadoks.

**Investigation:** Chiara Crestani, Ruth N. Zadoks.

**Methodology:** Chiara Crestani, John Bell, Samantha J. Lycett, Michael Fontaine, Ruth N. Zadoks.

**Project administration:** Chiara Crestani, Ruth N. Zadoks.

**Resources:** Laura M. A. Oliveira, Tatiana C. A. Pinto, Claudia G. Cobo-Ángel, Alejandro Ceballos-Márquez, Nguyen N. Phuoc, Wanna Sirimanapong, Swaine L. Chen, Dorota Jamrozy, Stephen D. Bentley, Ruth N. Zadoks.

**Supervision:** Taya L. Forde, Dorota Jamrozy, Michael Fontaine, Ruth N. Zadoks.

**Validation:** Chiara Crestani, Ruth N. Zadoks.

**Visualization:** Chiara Crestani.

**Writing – original draft:** Chiara Crestani.

**Writing – review & editing:** Chiara Crestani, Taya L. Forde, Ruth N. Zadoks.

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
