## [Decision Letter · Decision Letter 0]

12 Jun 2024

Dear Dr Crestani,

Thank you very much for submitting your manuscript "Genomic and functional determinants of host spectrum in Group B Streptococcus" for consideration at PLOS Pathogens. As with all papers reviewed by the journal, your manuscript was reviewed by members of the editorial board and by independent reviewers. In light of the reviews (below this email), we would like to invite the resubmission of a revised version that takes into account the reviewers' comments. Your revised manuscript may be sent to reviewers for further evaluation.

Sincerely,

Lakshmi Rajagopal, Ph.D

Academic Editor

PLOS Pathogens

Michael Wessels

Section Editor

PLOS Pathogens

Michael Malim

Editor-in-Chief

PLOS Pathogens

orcid.org/0000-0002-7699-2064

Reviewer's Responses to Questions

**Part I - Summary**

Reviewer #1: Chiara Crestani and colleagues described the global GBS population comprises host-generalist, host-adapted and host-restricted sublineages, as they named, through bioinformatics analysis by downloading the whole genome sequence data from public repositories or self-generated. This is an interesting idea.

However, the manuscript seemed to lack novelty. The evidence in support of the authors’ statement is insufficient. Furthermore, we think the manuscript was not well-organized.

For example, In line 2.5. In vivo assessment of the role of Locus 3 in fish infection.

Why the Lac.2 operon (GBS in cattle) and the scpB transposon (GBS in humans) were not selected for in-vivo experiment. The author should explain this in the introduction.

Reviewer #2: In this work, Crestani et al., analyze a robust set of 1,254 high-quality GBS genomes to identify potential drivers of host specificity. They identify three functional groups that describe host specificity – host generalists, host-adapted, and host restricted and assign GBS clonal groups (CGs) within these classes to the host species. They then identify genetic drivers that contribute to host-adaptation reiterating the association of scpB acquisition with human interactions and the Lac.2 operon with cattle. Interestingly they identify a novel genetic locus they termed Locus 3 that is associated with adaptation to fish hosts, and they show loss of Locus 3 significantly decreases GBS virulence in two distinct strain backgrounds and animal models. The findings presented confirm previous claims about scpB and Lac.2 while also contributing novel findings for GBS evolution in the context of fish hosts.

**Part II – Major Issues: Key Experiments Required for Acceptance**

Reviewer #1: (No Response)

Reviewer #2: (No Response)

**Part III – Minor Issues: Editorial and Data Presentation Modifications**

Reviewer #1: (No Response)

Reviewer #2: - Assuming mobile genetic element but ‘MGE’ should be defined in Ln 30. Spelled out in Ln 208

- Explanation of how 80% was selected as the cutoff for host generalist v host specialists would be helpful.

- Fig 2B could benefit from grouping the CGs by host generalist, host-adapted, and host restricted rather than numberical order across the X-axis

- Ln 168-169: Are the authors able to indicate where SL61 falls on Fig. 2C?

- Could the authors add labels or comment on the enriched human-associated unitigs that fall between 0.50 and 0.75 in the reference genome position in Fig 4A, especially the peak ~0.6.

- It is difficult to determine where the supplemental data citations are referencing. The tables are not cited in the text. In Table S2, what do the colors indicate? Are there corresponding legends to the supplemental data?

- The key for Fig. 6A says ‘Viet Nam’

- Ln 304-308: Are the colors representing Brazil and Vietnam inverted in this figure? The text cites that the emergence of the CG283 strains in Brazil occurred around 2015 but the figure looks as if this is the Vietnam strain emergence. And vice versa for the 2003 scpB loss that, according to the figure, appears to be isolated from strains in Brazil, the text cites Vietnam.

PLOS authors have the option to publish the peer review history of their article (what does this mean?). If published, this will include your full peer review and any attached files.

Reviewer #1: No

Reviewer #2: No
---

## [Editor Report · Decision Letter 1]

8 Jul 2024

Dear Dr Crestani,

We are pleased to inform you that your manuscript 'Genomic and functional determinants of host spectrum in Group B Streptococcus' has been provisionally accepted for publication in PLOS Pathogens.

Best regards,

Lakshmi Rajagopal, Ph.D

Academic Editor

PLOS Pathogens

Michael Wessels

Section Editor

PLOS Pathogens

Michael Malim

Editor-in-Chief

PLOS Pathogens

orcid.org/0000-0002-7699-2064
---

## [Editor Report · Acceptance letter]

7 Aug 2024

Dear Dr Crestani,

We are delighted to inform you that your manuscript, "Genomic and functional determinants of host spectrum in Group B Streptococcus," has been formally accepted for publication in PLOS Pathogens.

Best regards,

Michael Malim

Editor-in-Chief

PLOS Pathogens

orcid.org/0000-0002-7699-2064